# Exploration of Telemidwifery: An Initiation of Application Menu in Indonesia

**DOI:** 10.3390/ijerph191710713

**Published:** 2022-08-28

**Authors:** Alyxia Gita Stellata, Fedri Ruluwedrata Rinawan, Gatot Nyarumenteng Adhipurnawan Winarno, Ari Indra Susanti, Wanda Gusdya Purnama

**Affiliations:** 1Master of Midwifery Study Program, Faculty of Medicine, Universitas Padjadjaran, Jl. Eyckman No. 38, Bandung 40161, Indonesia; 2Department of Public Health, Faculty of Medicine, Universitas Padjadjaran, Jalan Ir. Soekarno KM. 21, Jatinangor, Sumedang 45363, Indonesia; 3Center for Health System Study and Health Workforce Education Innovation, Faculty of Medicine, Universitas Padjadjaran, Jl. Eyckman No. 38, Bandung 40161, Indonesia; 4Indonesian Society for Remote Sensing Branch West Java, Gedung 2, Fakultas Perikanan dan Ilmu Kelautan Universitas Padjadjaran, Jl. Ir. Soekarno KM. 21, Sumedang 45363, Indonesia; 5Department of Obstetrics and Gynecology, Faculty of Medicine, Universitas Padjadjaran, Bandung 45363, Indonesia; 6Hasan Sadikin Hospital Bandung, Bandung 40161, Indonesia; 7Mother and Child Health Division, Department of Public Health, Faculty of Medicine, Universitas Padjadjaran, Jl. Eyckman No. 38, Bandung 40161, Indonesia; 8Informatics Engineering Study Program, Faculty of Engineering, Universitas Pasundan, Jl. Dr. Setiabudi No.193, Bandung 40153, Indonesia

**Keywords:** telemidwifery, digital health, digital communication

## Abstract

The midwifery continuity-of-care model improves the quality and safety of midwifery services and is highly dependent on the quality of communication and information. The service uses a semi-automated chatbot-based digital health media service defined with the new term “telemidwifery”. This study aimed to explore the telemidwifery menu content for village midwives and pregnant women in the Purwakarta Regency, West Java, Indonesia. The qualitative research method was used to explore with focus group discussion (FGD). The data collection technique was purposive sampling. The research subjects were 15 village midwives and 6 multiparous pregnant women. The results of this study involved 15 characteristics of menu content: (1) Naming, (2) Digital Communication, (3) Digital Health Services, (4) Telemidwifery Features, (5) Digital Check Features, (6) Media Services, (7) Attractiveness, (8) Display, (9) Ease of Use, (10) Clarity of Instructions, (11) Use of Language, (12) Substances, (13) Benefits, (14) Appropriateness of Values, and (15) Supporting Components. The content characteristics of this telemidwifery menu were assigned to the ISO 9126 Model standards for usability, functionality, and efficiency. The conclusion is that the 15 themes constitute the characteristic menu content required within the initiation of telemidwifery.

## 1. Introduction

There was an increase in maternal deaths in 2020 in Indonesia, up to 4627 cases from 4221 cases in 2019 [1]. In 2020, the number of infant deaths was 28,158, a decrease from 2019, which was 29,322 [1,2]. The province with the highest maternal (16.20%) and infant (10.45%) mortality in 2020 was West Java, with Purwakarta Regency being in the top ten regions with the highest maternal and infant deaths, with 33 and 72 cases, respectively [1,2,3,4]. This prevalence of maternal and infant mortality is indirectly caused by several factors, including the limited availability of health workers in basic service facilities, community knowledge, support from various sectors, and health facilities. The provision of integrated healthcare centers or *Pos Pelayanan Terpadu* (*Posyandu*) facilities is not evenly distributed and not entirely accessible to the population [4,5]. The *Posyandu* is a form of community-based empowerment to improve community health [6]. One of the criteria of active *posyandu* consists of having a mother and child health (MCH) program, including nutrition, immunization, and family planning services, which accounts for 50% of minimum coverage [7,8,9]. In 2020, there were 108 of 514 districts/cities (21.0%) in 15 provinces in Indonesia with active *Posyandu.* Active *Posyandu* is *Posyandu* with a minimum of 80% activity. West Java Province has one of the three lowest rankings (25.9%) of active *P**osyandu* [3]. The ratio of *Posyandu* to the number of villages in the Purwakarta Regency was 18.60%. This district experienced an increased use in at least 80% of *P**osyandu* from 2018 to 2020: 59.86%, 60.6%, and 99.2%, respectively [6,10]. Then, the use of the iPosyandu application (app), an initiative in the Purwakarta Regency in 2017, increased the number of active *Posyandu* in 2020 [7,9]. The initiative aims to optimize the role of *Posyandu* by using applications and websites through various developments to overcome limited access to health facilities [7].

Limited healthcare facilities cause an increase in the maternal mortality rate (MMR). This problem is also reinforced by the “3Ds” (delay in going to facilities, delay in getting referrals, delay in making decisions) factor, especially delays in reaching healthcare facilities, which cause slow and imprecise referral services [11]. Society considers communication difficulties as the cause of most medical accidents. Poor communication often causes delayed responses to consultations or referrals, diagnoses, and treatment [12]. Health information is also needed to reduce unspecific information and to enable people to make informed health decisions [13]. The health workers who play a major role in eradicating MMR and infant mortality rate (IMR) are midwives. Midwives will find it easier to establish a trusting relationship with clients if they have therapeutic communication skills, which helps them to be more effective in achieving the goals of applied midwifery care and providing professional services [12]. Care must be continuous, tasteful, and not authoritarian, and must respect women’s choices about where to give birth [14]. The delivery of health information is closely related to the effectiveness and efficiency of communication [12]. Thus, there is an essential need to improve the quality of communication in handling serious illnesses through communication skills, tools, patient education, and midwifery care models, one of which is the Continuity of Care (COC) model of midwifery care [15,16].

Previous research pointed out that women who receive continuous care are seven times happier if assisted by a known midwife and have a reduced risk of miscarriage (16%) and a reduced risk of miscarriage before 24 weeks (19%). Moreover, they were able to concentrate on regional anesthesia (15%), reduced the risk of preterm (24%), and reduced the risk of preterm birth and deliveries by episiotomy (16%) [17]. Other studies, in particular, suggest that communication, interdisciplinary familial relationships, and managerial support are critical to the sustainability and significant improvement of continuity in midwifery case care [16]. Therefore, the success or failure of implementing continuity of care depends on quality communication and information about midwifery services. This study will be used to create a digital health-based media service. Digital health involves using information and communication technologies to improve health and care outcomes [18]. Digital health services include eHealth, telemedicine, telemonitoring, and digital therapy, which are remote medical services [19,20,21,22,23,24]. Several articles report good results regarding the benefits of digital health using web platforms and mHealth to improve midwifery services, such as research on clinical decision-making support systems (mCDMSS), Continuum of Care Services (CCS)—Mother and Child Health (MCH), MAMAACT, iPosyandu, The COMmunity homebased INDia (COMB-IND), Roadmap, and Interactive Mobile massaging Alert System (IMAS) [24,25,26,27,28,29,30]. Referral rates are higher when access to the mHealth platform is readily available, improving complication detection and timely referrals to treatment facilities [31].

In this current study, Midwifery Continuity of Care (MCOC) is a continuous development of the iPosyandu application model that has existed since 2017 and started in Pasawahan District, Purwakarta Regency. This application initially focused on increasing community health workers’/*Kaders’* knowledge to use applications to overcome existing problems, which in their development were also intended for parents in monitoring their child’s growth and development [7,32]. However, there are still many obstacles regarding the concept and the understanding of village midwives in Purwakarta Regency in implementing midwifery continuity of care. Due to geographical differences and patient characteristics between rural and urban areas, it is difficult to monitor, collect, and enforce continuity of care with the same midwife. Based on this indicator, in subsequent developments, iPosyandu can be improved by adding the iPosyandu for a midwife (*Bidan*) named iPosyandu Bidan application, which will be enriched with MCOC-based menus or midwifery continuity of care under the name telemidwifery (Appendix A). The term telemidwifery, focused on midwifery continuity of care, has not been found, especially in Indonesia, based on keyword discovery in a recent literature review [33] or in other research [24,25,26,27,28,29,30]. Thus, the initiation of the application menu in terms of telemidwifery with sophisticated features is expected to make it easier for village midwives to gather patients to carry out continuity of care using the application (app) on a smartphone. However, in the process of creating a new application menu, it must be compared with standard theory to determine the suitability of new menu content designed through a qualitative process, which is the ISO/IEC 9126 model theory of software quality. Software quality models and their attributes may be used in many contexts, for instance, during the development of a new application, especially a telemidwifery menu in the iPosyandu Bidan application [34].

This research falls under the research umbrella of iPosyandu as the initiation of a telemidwifery menu in the iPosyandu Bidan application. This menu will be implemented using a semi-automated system with a chatbot or manual digital communication reference to implement midwifery consultations in remote care [18,35,36,37]. Thus, this menu application proposes a sophisticated feature to avoid the previous weakness of digital health applications [30]. Researchers aimed to explore (1) the initiation of telemidwifery menu content including its benefits, and (2) perceptions, expectations, and points of view of the village midwives and pregnant women to succeed in designing menu content that improves the application.

## 2. Materials and Methods

### 2.1. Study Design

In this study, a qualitative method was used to explore the characteristics of telemidwifery content as the basis for the continuing development of digital communication in the iPosyandu Bidan. The qualitative theoretical approach is an exploratory qualitative analysis in which qualitative researchers follow a series of steps in exploring new areas of social or psychological life by collecting open data to produce new concepts and generalizations [38].

### 2.2. Recruitment and Participants

The population in this qualitative study included 211 village midwives spread over 20 working areas at the primary level of public health centers, or *Pusat Kesehatan Masyarakat* (*Puskesmas*), in Purwakarta Regency. From the population, we took a sample using the purposive theoretical sampling technique. The inclusion criteria were: (1) Village midwives who work in the working area of the *Puskesmas* with a high and a low number of maternal and infant deaths in regions with and without wireless signals, who used and did not use the iPosyandu; (2) the village midwife had a history of midwifery education, (3) attended iPosyandu training, and (4) had more than a 3-year working period. Through the first inclusion criterion, three *P**uskesmas* (*Puskesmas* Kiarapedes, *Puskesmas* Purwakarta, and *Puskesmas* Koncara) were found to be accessible populations, with a total of 22 village midwives. Then, based on the second, third, and fourth inclusion criteria, we found 15 eligible village midwives as respondents in a face-to-face focus group discussion (FGD). The recommended range of respondents for a qualitative study is 6–12 persons [39].

The population of pregnant women was obtained from the following inclusion criteria: Pregnant women in 1st–3rd trimesters; multiparous pregnant women; pregnant women in the working area of *Puskesmas* who attended iPosyandu application training. The only *Puskesmas* that met the fourth inclusion criteria was the *Puskesmas* Pasawahan, with 899 pregnant women. Then, an accessible population of 11 pregnant women came to the first iPosyandu application training for pregnant women on 15 March 2020. Thus, from eleven people, six pregnant women were eligible to be included in the FGD because they fulfilled the qualification of first and second inclusion criteria. The two groups of respondents are related in this research to explore expectations, preferences, and points of view—village midwives as service providers and multiparous pregnant women as recipients of continuity midwifery services. The village midwife provides information related to pregnancy, preparation for delivery, the postpartum period, care for infants and toddlers, and the use of family planning contraception. Their opinions will enrich the initial menu content for telemidwifery and complement each other.

### 2.3. Data Collection

In the FGD, the research instrument was the researcher equipped with tools in the form of an audio recorder, camera, field notes, and interview guidelines [40]. Qualitative data collection was carried out from December 2021 to March 2022. The discussion process was carried out within a time range of 60–90 min. The researcher used data sources, methods, theory, and researcher triangulation to minimize the bias. Researcher triangulation was conducted because researchers belonged to the iPosyandu umbrella, a multidisciplinary profession. Hence, in the FGD process, the researchers worked as a team to succeed in the discussion, in roles such as the moderator (to guide the discussion panels), notetaker, observer, and facilitator (to lead the conversation process (first author)).

To maintain data source triangulation, these FGDs were conducted with participants, village midwives, and pregnant women. The aims of the FGD—to find out their perceptions and expectations to explore the content of a sustainable midwifery service menu based on semi-automatic chatbots in telemidwifery—were explained to the respondents before starting the discussion. The researcher used a structured-question guide form for general topics (Appendix B) with a stated key probing list. The key probing list was discovered based on scoping review results.

Hence, from various theories of ISO/IEC’s software engineering to conduct theory triangulation, the theoretical propositions adopted the attribute theory of software quality ISO/IEC 9126 Model (usability, functionality, efficiency, reliability, portability, and maintainability) to design and develop the new application [41,42]. The usability and functionality attribute findings, which address technical issues in telemidwifery initiation, are derived from conceptual variables in propositions [43,44]. The theoretical proposition was gained from the literature and previous research. The theoretical propositions of this research included naming, appearance, substance, attractiveness, ease of use, benefits, clarity of instructions, language, and suitability of values as the core contents of the telemidwifery menu.

### 2.4. Data Analysis

To analyze the content of the telemidwifery menu, the NVivo Release 1.6.1 (1137) software QSR International (Burlington, MA, USA) was used. This stage of analysis refers to the opinion expressed by Miles and Huberman. Three phases must be carried out in analyzing qualitative research data: (1) data condensation, (2) data display, and (3) conclusion drawing/verification. Researchers used the data’s validity to confirm the study’s truth. The validity criteria consist of credibility, transferability, dependability, and confirmability [45].

### 2.5. Research Ethics

The research was conducted after obtaining approval from the health research ethics committee of the Faculty of Medicine, Universitas Padjadjaran with the number: 232/UN6.KEP/EC/2022. This research is part of the iPosyandu research umbrella, with the approval of the ethics committee number: 640/UN6.KEP/EC/2021.

## 3. Results

The qualitative study conducted through FGDs resulted in a great deal of input on menu content for initiating the telemidwifery menu in the development of iPosyandu Bidan. The results of the data condensation include the coding process, the category and theme determination results obtained from 213 codings, 59 categories, and 15 major themes as illustrated in Figure 1. Thus, the detailed visualization findings of 15 major themes are shown in Figure 2.

### 3.1. Categories of Description

The findings regarding themes are supported by the categorization and coding (key insight) as shown in Table 1, which provides a review of expectations, preferences, and points of view from respondents in the focus group discussion. The themes and key insights were gained from conversation between respondents and researchers. The conversation yielded quotations that enriched the idea of creating menu content for telemidwifery. Table 1 presents the chosen quotes that represent key insights as an overview of the source of organized telemidwifery menu content (themes) findings.

#### 3.1.1. Naming

Several village midwives stated that the name “Telemidwifery” was suitable for this menu. Meanwhile, in extracting these data, two observations were also made that the term telemidwifery was able to present the purpose of the menu to carry out midwifery continuity-of-care services. This name also represented the midwife’s identity as the primary profession in midwifery continuity-of-care services. The village midwives stated that this also indicated a naming for the application menu owned by the midwife that differed from the existing application naming.

#### 3.1.2. Digital Communication

All village midwives wanted an application menu that could answer directly (automatically) based on chatbots to avoid long responses from midwives when giving manual answers so that the waiting time for patients to receive responses did not take as long as in some cases. This was in line with the pregnant women’s opinions, who also wanted an application menu that contained digital communications that could automatically answer. However, pregnant women also wanted a combination with manual digital communication to receive more precise answers. If they are not entirely satisfied with answers from the chatbot, they can communicate directly to submit complaints via chat and telephone.

#### 3.1.3. Digital Health Services

The digital health services theme contains the same categories as the digital communication theme, which are manual and automatic. Thus, the different items in this category pertained to functions to use and operate the menu. Digital health service focuses on how the midwife responds to the consultation and checks the patient’s condition in remote care services. It uses a sophisticated feature (telehealth; pulse, blood pressure, breath, temperature, contractions, etc., using an application sensor in accordance with the principle of remote care) of telemidwifery. However, the category of digital communication theme focuses on how to communicate with the patient aside from medical examinations.

The village midwives stated that all of them wanted an automatic, digital midwifery health service to provide many quality services at once. Most midwives also thought that chatbots’ automatic answers about midwifery materials should be physiological or minor complaints. It is intended that midwives can provide fast midwifery services to patients to increase patient satisfaction in midwifery services based on the telemidwifery application menu.

The same thing was stated in pregnant women’s opinions: they also wanted minor tips about midwifery material to be answered automatically by the chatbot. However, they assumed that the services provided by this robot could not produce valid examination data because the conditions that the chatbot could provide answers for did not necessarily match the state complained of by the patient. The village midwife also declared a solution that there must be a combination of automatic and manual services called semi-automated. Several midwives stated that for services that require direct examination, it is better to manually consult directly with the midwife via an application or face-to-face meetings, especially for answering questions and services outside the capacity of the chatbot. As for pathological obstetrics material, it is better to ask the midwife manually because it is an emergency and a dangerous condition. Pregnant women also wanted manual digital health services via chatbots’ application menu for automated midwifery services, which felt unclear. Moreover, pregnant women were still worried that this menu cannot quickly answer every complaint submitted. Hence, they would choose to call or communicate directly with the midwife via customer service.

Based on the expectations of the village midwife, questions that the midwife wants to answer manually must adjust to the midwife’s busyness so that the midwife can provide a choice of the longest waiting time. The respondents recommended 30 min, and the midwife can give a choice of time, which can be clicked automatically so that the patient can know that they have to wait a moment, according to the time suggested by the village midwife. However, pregnant women stated that they should not wait too long for an answer or for manual midwife services to be able to answer their curiosity quickly. Thus, they want the longest waiting time to be around 10–15 min.

#### 3.1.4. Features of Telemidwifery

The village midwife expected a feature in this menu to be another menu link. Another menu link could be a link that connects users to other menus, such as a doctor’s consult menu and a referral menu outside the telemidwifery menu. This feature is needed because village midwives often receive questions or midwifery cases that are beyond their authority. The pregnant women also wanted a link to the educational video menu to obtain complete information. Another feature expected in this menu is the notification feature. According to the village midwife, this feature is expected to be able to inform them that there is an incoming question or service that the midwife wants to answer directly. This feature is also likely to appear and be answered automatically without first opening the menu application. The midwife can respond more quickly in answering and avoid forgetting to reply. 

In addition to notifications, another feature that is expected in the menu is a reminder feature. This feature is also likely to be equipped with an alarm. Thus, respondents expected to be reminded one day before and after scheduled appointments. However, this is slightly different from the results of the opinions of pregnant women. They wanted to be reminded a week before, three days before, and the day before the scheduled appointment. Therefore, the reminder feature is also related to the rescheduling feature, which functions to rearrange the missed schedule and if there are sudden activities from the village midwife. In line with village midwives, pregnant women also wanted a rescheduling menu in case they forget and miss a scheduled appointment that has been set. Another feature that the village midwife expected is a translator feature or automatic translation in the menu. This feature is expected because many midwives are still constrained by English content. The village midwife also expected a zoom feature to enlarge text, images, and videos automatically according to user needs.

#### 3.1.5. Digital Check Features

In discussing the digital check feature with the village midwife, it was found that they expected a small digital sensor checking tool capable of performing automatic services. For example, this tool could be used for the examination of the anamnesis: the first day of the last menstrual cycle (FDLMC), body mass index (BMI), tetanus (TT) injection; vital signs: blood pressure, pulse, respiration, oxygen levels, fetal heart rate (FHR), and temperature; physical check: mid-upper arm circumference (MUAC), body weight, body height, fetal movement, and Leopold; and supporting examinations: Hemoglobin (Hb) level. In line with pregnant women’s expectations, especially in the FHR examination, they can be more reflective about the condition of the fetus they are carrying.

Village midwives raised concerns about the accuracy of the digital sensor embedded in this menu. They believed there could be a discrepancy between checking or measurement with the application menu and direct measurements, so a review of the checking results would be needed to confirm that they are correct. The village midwives also suggested that the accuracy of this digital sensor should be tested before being used in the telemidwifery menu and disseminated to the general public.

#### 3.1.6. Service Media

The village midwives expected service media in the form of written media (chat), visual media (sending photos and pictures), audio media (voice notes), and audio–visual media (video calls and sending videos). Media sent as photos, images, and videos are expected to help midwives to carry out an initial examination in the form of checking to confirm the truth of the complaints submitted by patients. Videos are also expected to provide answers accompanied by examples so that patients can practice independently at home. A video can also explain things that are considered taboo, such as complaints in sensitive areas. As for the voice note feature, village midwives and pregnant women both wanted this feature to be able to submit complaints and explain answers to long questions without the need of typing. Some said that it is better to call if they have trouble typing. Regarding video call media, compared to the number who agreed, most village midwives objected to this feature because it was considered to cause inconvenience in the form of disturbing private space as well as the tendency of patients to continue making video calls without estimating the time.

Different results were obtained from the pregnant women’s group; they expected a video call feature that would make it easy and comfortable for them to submit complaints because they could chat face-to-face directly via the application with their midwife. Likewise, with telephones, some midwives also stated that patients tend to keep repeatedly calling without knowing the time if the call is answered only once. Contrary to the pregnant women’s wish, they wanted a telephone feature to submit complaints directly without requiring extended typing. Finally, in the FGD with the village midwives, a solution was found that the village midwife wanted an option in the form of an automatic notification that would appear when the patient wanted to be serviced manually.

#### 3.1.7. Attraction

In this dimension, the village midwives said that things that can increase the attractiveness of using this menu are an attractive appearance and features, the credibility of the material, completeness of telemidwifery features, and visualization, especially menus that can present communicative and interactive content. The telemidwifery feature here is related to Digital Communication and Digital Health Services, which can be accessed semi-automatically. This was in line with the pregnant women’s hope, who also wanted content visualization in the form of graphs of their baby’s growth and development levels. The village midwife also stated that menu and video content with animated images, phantoms, and graphic displays would increase the attractiveness of the menu for the village midwife. From the patient’s perspective, the font type used in this menu will also increase interest in using it, especially for content that is new, so it does not seem stiff. Pregnant women also want service features that can be answered directly by robots.

#### 3.1.8. Display

Regarding the display of telemidwifery, respondents discussed several indicators in the form of menu icons, display colors, menu writing, writing colors, and writing markers. For the menu icon, the midwives stated that the menu icon with pictures of pregnant women, babies, and children was boring even though obstetrics were usually associated with these icons. Some began to understand the essence of this menu, that the midwife is the leading actor and holds the application menu, so they also suggested icons that focus on midwives, midwives with hairnets, midwives with stethoscopes, and midwives with cell phones or telephones according to the phrase “tele”. The icon is expected to represent the menu’s purpose and the midwife’s identity. An attractive icon makes the application easy to remember. The village midwife said they wanted bright but soft colors and more pastels regarding the display color. They wanted the dominant color to be pink because the midwife is associated with the color pink, and the pomegranate (Indonesian Midwifery Association, IMA logo) and the MCH book are also pink. However, they suggest other colors with pastel shades that are not monotonous: purple, lilac, white, dusty pink, yellow, orange, blue, turquoise, and mint green.

For the writing on this menu, the midwives of several villages suggested only standard types of writing, such as Times New Roman and Calibri, but adjusted to the current trend of fonts; they suggested avoiding the form of cursive writing so that it is easy and clear to read. The size of the script was also discussed, with a preference for a medium size that is not too small for easy reading. Some suggested that the size could be adjusted according to the user’s wishes, which can be changed by themselves with the zoom (magnifier) feature. For the color of the writing, the village midwife suggested using a black script instead of a colorful one because they were afraid that it would contrast with the background color so that the handwriting was not legible. Some pregnant women also stated that they wanted black writing and were given markers for writing essential words in red or blue color to be visible. Similar to the results of the FGD with village midwives, they also wanted writing markers for actual words in red, italicized, and bold.

#### 3.1.9. Ease of Use

The village midwife stated that a telemidwifery feature was needed to bring users convenience. To lighten the workload of midwives, automatic features based on semi-automatic functionality are easy to operate, do not complicate the service, and are practical and sophisticated. The ease of use of this menu can also be supported by the ability to access it. The access is up to 24 h for automated chatbots and almost 12 h, according to the midwife’s operating hours, for manual services.

#### 3.1.10. Clarity of Instruction

The clarity of instructions is obtained through understandable language, suitability of submenu symbols, and service preference indicators. In operating the menu, it is hoped that the instructions will be in Indonesian because this menu will not only be used within the Purwakarta Regency. It can even go abroad, so it must also be equipped with a bilingual sub-choice. Likewise, pregnant women’s preference showed that Indonesian language instructions would be easier to understand for more people. The submenu conformity indicator is expected to be in line between the symbols and the executed functions. It aims to make it easier for the midwife to select the menu or command to be performed according to the midwife’s preferences. Pregnant women stated that if a submenu can be directly clicked, it will make it easier for them to receive and enter instructions without having to spend a long time searching and typing the command. The suitability of a submenu is also related to service preferences that can be adjusted to the condition of the midwife regarding their choice to receive services with specific features to avoid discomfort and lengthy patient waiting times.

#### 3.1.11. Use of Language

In the discussion of the use of language, three indicators were found. The indicators are the type of language, the delivery method, and the greeting. In this discussion, village midwives and pregnant mothers wanted this menu to be in Indonesian because it is the national language, and it could be used in all of Indonesia. However, several village midwives argued that it is also necessary to include English for use by a foreign patient who wants to receive continuity midwifery services in Indonesia. Thus, the village midwife suggested using two languages, English and Indonesian, and almost all of them refused to use the regional language because the speech could be misinterpreted by people who were not native speakers. However, some village midwives also stated that it is okay to use the local language, if possible, for manual services.

Regarding the delivery method, village midwives and pregnant mothers wanted the delivery to be formal and informal. Although it is stated that the formal variety seems stiff, for the language of customer service, it is customary to use formal language to make it seem more serious and polite. The use of the formal variety does seem more familiar and communicative. According to some village midwives, affixing emoticons can lead to misinterpretation by the user and even the impression of ridicule. Then, a solution was found that formal language should be intended for services performed by chatbots, and informal language should be intended for services carried out manually by midwives. Thus, midwifery services are welcome to use a variety of formal and informal languages. The variety of language depends on the intensity of service and the closeness between the patient and the midwife. For greeting words, several village midwives stated that, to avoid misperceptions and prioritizing one party, there was a need for greeting words that are usually only used as general greetings because Indonesian people are multicultural. Therefore, it is better to use standard greetings such as “hi” and “hello”.

#### 3.1.12. Substance

According to the group discussion results, the theme of substance consists of several indicators: the importance of care for pregnancy, childbirth, postpartum, infants and toddlers, and family planning. All substantive material is needed and recommended in all parts of a woman’s life cycle, which village midwives and pregnant women all agreed on. They wanted questions about pregnancy, delivery, postpartum, infants and toddlers, as well as family planning care that can be answered automatically by the chatbot.

In the reference source indicator, all village midwives stated that the reference sources can guarantee the validity, credibility, and quality of materials, that they are based on the latest evidence-based care based on government regulations, health and midwifery books, mother and children health (MCH) books, research journal articles, health organizations, or professional organizations such as the World Health Organization (WHO) and the Indonesian Midwife Association (IMA). The village midwife strongly recommended relying on the MCH book because the patient also holds the MCH book, so that the patient can read not only the menu but also refer to the book itself; this recommendation is supported by the fact that the MCH book has a national standard, and its content is updated regularly.

#### 3.1.13. Benefits

The results of the FGD on the theme of benefits obtained several supporting indicators, including service optimization, changes in habits, changes in behavior, and changes in attitudes. Service optimization indicators include early detection of complications, service effectiveness, time efficiency, speed of service, suitability of authority, punctuality of examinations, reducing costs, reducing disease transmission, reducing waiting time for services, and responding to emergencies. This result also emerged from the opinion of several village midwives regarding the expected benefits of early complication detection, which is a continuation of emergency response actions that can have implications for increasing the optimization of midwifery services.

Meanwhile, indicators of habit change include the intensity of using the menu, reducing the belief in myths, and increasing openness about sensitive matters when consulting midwives via the menu. Indicators of behavior change relate to improving health and increasing knowledge. We all know that the aspects of behavior change have cognitive, affective, and psychomotor aspects. Therefore, on this menu, it is hoped that there will be behavioral changes in the cognitive aspect. Supported by indicators of social change, this menu is expected to improve interaction and communication patterns between midwives and their patients, which are more intimate and warmer, to increase closeness that leads to comfort and trust in consulting and receiving services via the menu.

#### 3.1.14. Appropriateness of Values

Based on the discussion with the village midwife, it was found that the theme of value conformity was divided into two indicators: social values and cultural values. The expected social value is related to the intensity and frequency of interaction between midwives and patients through the application menu. This interaction can develop the quality of a closer relationship because it follows the phases of the patient’s life from pregnancy to family planning. The cultural values are extracted from the discussion of all village midwives regarding the myths still firmly held by patients, so it is hoped that this menu will provide better knowledge and be easy to understand without offending the customs and beliefs of the local community.

#### 3.1.15. Supporting Components

There was a theme regarding supporting components, which were endorsed by indicators of data size, operating hours, signal requirements, and menu operators. The village midwives strongly emphasized the compatibility of the operating hours of the manual menu service with the midwife’s schedule of activities in the hope of avoiding inconvenience between midwives who were disturbed because they had other business and patients who could not wait for a lengthy response. It is intended that the output of this menu can be maximally successful. The signal requirement indicator is considered to support optimizing menu use because regional and weather conditions influence the presence or absence of a signal and the data operator used. This result is also in line with several pregnant women’s opinions that the data network often disappears, especially when it rains, and that data operators are only good in specific areas. Some village midwives and pregnant women also suggested that this menu would be better if it were handled by the admin or customer service—a kind of automatic operator that can respond first before the midwife answers, and more importantly, according to pregnant women, is a menu operator who can be called at any time.

#### 3.1.16. ISO/IEC 9126. Attribute Finding

Based on the qualitative results of discussions with village midwives and pregnant women, a conceptual framework model for the content of the telemidwifery menu can be made whose software quality was reviewed based on “The ISO/IEC 9126 Model” as illustrated in Figure 3.

The author grouped and categorized the sub-characteristics and content concepts of the telemidwifery menu with qualitative results, starting from six common characteristics that describe software quality requirements. As stated in the standard ISO 9126 model, software quality can be evaluated and designed with the following characteristics: usability, functionality, reliability, efficiency, portability, and maintainability. These high-level characteristics provide a conceptual basis for further refinement and description of quality. However, the relative importance of each element in quality requirements varies depending on the user’s point of view and the application domain under consideration. The ISO standard defines three quality views: a user view, a developer view, and a manager view. In particular, the opinions of users in this study are those of village midwives (users in iPosyandu Bidan) and pregnant women (users in iPosyandu for family for future development). Therefore, users pay attention to using the site, its performance, its search and browsing functions, user-oriented unique content and functionality, its reliability, feedback, and aesthetic features, and finally, they are interested in the quality of its use. However, maintainability and portability are not what users discovered.

In this study, researchers found three characteristics of software development from the qualitative results: usability, functionality, and efficiency. The concept of reliability was not found in the qualitative results because the qualitative research explored the expectations, hopes, desires, and preferences of village midwives and pregnant women, that is, regarding the expected new application menu apart from the implementation of the application menu trial and not in the form of an evaluation test. According to the Informatics Expert, these three characteristics are sufficient for the initial development component of the new application menu. Usability is the ease with which software can be understood, learned, used, configured, and executed when used under specific conditions. This study’s usability subcategories included menu naming, telemidwifery features, clarity of instructions, ease of use, appearance, attractiveness, substance, use of language, suitability of values, service media, and benefits. Functionality is described in terms of attributes that define the existence of a particular group of functions and specified properties, which in this case are subcategories of digital communication, digital health services, and digital service features. Meanwhile, there are subcategories that are shared between usability and functionality: service media and benefits. Efficiency is related to attributes that describe software performance concerning resource and time utilization. Subcategories related to efficiency in this study are supporting components that contain operational hours, signal conditions, data quantities, and menu operators.

## 4. Discussion

(1)Naming

The term *telemidwifery* appeared in 2002 as “*telemidwifery calls*”, which are only specifically for breastfeeding counseling services without any specific explanation of the function [46]. The title or name of the application is designed to direct users to feel at ease when seeking health information and medical assistance for their prenatal and postnatal care [35]. The name that shows the service provider’s identity will better represent the purpose of the application menu [47]. Making a name for any product must take into account its formation, meaning, pronunciation, related-to-product features, whether it is easy to pronounce and pleasant to hear, a short and straightforward morphology, and the views of community members because it affects the production of products with attractiveness to consumers [47,48]. Therefore, in forming a name for this new menu, the author first discussed with the *end-users*, in this case, the village midwife, to find out their interests, suitability of meaning, and purpose and to present the midwife as the owner.

(2)Digital Communication

A study applies examples of semi-automatic use in digital communication, a combination of automated and manual communication. Previous research carried out a semi-automated *chatbot-based application* that aims to provide automatic answer templates for the same question in detail, consistency, and quality. Then, if the system cannot read questions, they will be answered manually, comprehensively, and consistently by the instructor, which is a personalized approach while significantly saving the provider time [49]. Features such as *chatbots* can be used to enable patient–provider communication and peer support. The semi-automated short message intervention showed that automated messages encouraged greater engagement and inquiries from pregnant women to nurses, explored the semi-automated system of physician work in application forums, and could better support patients asking clinical questions [50]. This study explains that the first two communication processes occur via the application. If the midwife cannot handle it or there is still information lacking in detail, it can be carried out further through manual communication by making an appointment. With the automatic stage and then the manual step, patients assume the use of *digital communication* positively impacts their access to maternal and child health information because it can increase interaction and close relationships between midwives and patients so that more open communication is established [51,52,53].

(3)Digital Health Service

The adoption of mobile technology-based health services (*mHealth)* has made healthcare more accessible and affordable [22,23]. *Chatbots* are a key technology poised to transform healthcare shortly. *Chatbots* are proposed as automated or semi-automated therapeutic and diagnostic tools [54]. *The chatbot* will conduct a short and easy health survey, especially in an emergency; when looking for a particular sign or pattern that can be used to diagnose a disease, *the chatbot* can react quickly or immediately to questions related to the patient’s healthcare [35,55]. Several health professionals highlighted that some patients experience chat as “robots,” speculating that this could affect the relationship between patients and healthcare providers, so communicating via text causes a loss of communication nuance [35,56]. Thus, to support the success of *telemidwifery* as a semi-automatic *chatbot-based digital health application, it* must be integrated with manual services. Supposing the question cannot be concluded through digital communication, the patient can be scheduled for a relevant physical appointment with the health worker concerned or transferred to a health worker or medical personnel according to their work authorization [51].

However, challenges remain with application-based *digital health*, as one study stated that specific patients needed several hours to respond [51]. A study said there was dissatisfaction with using *chatbots* because of the long waiting time, which was around 1.20 min in replying. Therefore, the response time is an important thing to consider in making the application menu. The same study also stated that the application’s consultation session was limited to 10–30 min per session [57]. This is in line with the findings in the qualitative results that, in the manual service stage, village midwives and pregnant women wanted a response time of around 10–30 min so that they do not wait too long and eliminate interest in using *telemidwifery*.

(4)Telemidwifery Feature

The importance of these features in an application menu is also stated in previous research. It said that the improvement of application-based services cannot be separated from the completeness of the existing features in the application, two of which are the *reminder*, notification, or interactive message warning and the appointment scheduling feature used in the study. *The reminder feature* has proven effective in empowering pregnant women through greater access to information about crucial danger signs and birth preparedness accompanied by an alert sound [13,55,57,58]. However, alarms and *notifications* that are too frequent will be annoying and drain the patient’s cell phone battery. Patients prefer applications that can be customized according to user preferences [59].

In similar research, utilization of other menu link features, referred to as linked dialog blocks, helps handle unexpected responses from users and is an alternative solution for finding unwanted information from a pre-arranged *chatbot-based conversational programming flow* [35]. Other menu link features are also expected to facilitate the search for more specific information and patient handling under the authority of the relevant health workers, such as in the referral process. *mHealth* technology is feasible and can improve the detection of complications and timely referral to health facilities [31]. As for the translator feature, it was also found in other studies that researchers also need to make a dictionary in addition to preparing a question-and-answer knowledge database. In this case, it is a synonym dictionary to increase accuracy in providing question-and-answer pairs that are under the user’s intent (a variety of abbreviations for medical terms and neologisms) [35].

(5)Digital Check Features

Digital sensors for these qualitative findings include pulse, breathing, oximetry, blood pressure, fetal heart rate, and Leopold checkers, whose examinations can be carried out remotely via the application menu on the phone. In line with previous research, the patient’s motivation to use a digital sensor-based mobile pregnancy application is to receive information about fetal development and changes in a woman’s body during pregnancy, control weight gain, and monitor fetal development and changes in the body [60]. The previous smartphone platform developed for the Android system integrates peripheral sensor input devices in Guatemala. The device includes pulse oximetry (*Onyx II, Model 9560, Nonin Medical, Inc, Plymouth, MN, USA)* and a 1-dimensional Doppler handheld *ultrasound device* (*AngelSounds Fetal Doppler JPD-100 s, Jumper Medical Co., Ltd, Shenzhen, China*). The camera app is also adapted to a self-inflating oscillometric blood pressure cuff (*Omron M7, OMRON Healthcare, Kyoto, Japan)* [31]. A digital sensor in an application improves timely and accurate referrals to a higher level of care. Increased care, measured as a referral rate per number of births and as a proportion of successful referrals, can be achieved by increasing the diagnostic capabilities of participating TBAs and linking those with existing referral networks [31,60].

(6)Service Media

Audio media can be telephones, *voice notes,* or recordings containing specific information. Video and audio information make it easy for users to understand because it is a visual message that can be replayed and listened to as often as the user wants [61]. This type of service media is expected to facilitate village midwives’ ability to provide long-distance continuity midwifery services. *Telemidwifery is a subset of chatbot-based* digital healthcare technologies that promote intelligent, software-assisted service delivery that communicates with people in their native language via voice or text [55]. Receiving video and audio with information related to maternal health and the opportunity to call when needed is considered a friendly and adaptable source of maternal health information, given the nature of the video/audio, which is easy to understand and remember compared to verbal health talks that are often obtained in clinics [61]. In addition to audio media, visual media, and audio–visual media, written media is the main thing in this *telemidwifery menu* because it is based on a *chatbot in the form of a* question-and-answer *chat* page. *Chatbots*, in particular, typically support text-based conversations or clickable responses and are designed to look like instant messaging apps [24,35,54].

(7)Attractiveness

Digitization penetrates almost all areas of modern life because of the portability of the exclusive features of smartphones and tablets, with their increasingly sophisticated interactive applications that are automated, available at any time, and user-friendly, which can increase the attractiveness of applications [62,63]. The application’s attractiveness will also increase relative to complete features, one of which is transportation access [28]. In addition to the completeness of features, the credibility of the content of an application must be considered. A study says that the health content offered must be created and validated by specialized professionals and supported by research institutions [35,60,61]. An attractive appearance must also be considered when developing a new application menu. Creating an attractive interface (visualization) layout following information needs is necessary for designing a mobile application system.

(8)Appearance

A good menu application appearance or graphical user interface (GUI) is needed to bridge quality and user needs [7]. If the client is familiar with the interface of an application and it operates according to their needs, the application will become a comfortable platform and maintain privacy [13]. A unique and well-known product influences the essence of marketing and business promotion prospects. The logo or icon affects the promotion of the product used [47]. *Chatbots* will appear more human-like if an avatar image and firm tone of voice are added to make the conversation more relaxed [35,55].

(9)Ease of Use

Sophisticated digital technologies (e.g., mobile/digital apps, SMS/text messaging, and wearables), which in this case is the initiation *of telemidwifery menus*, can provide unprecedented opportunities to reach and engage the broader community in providing continuity midwifery services [62,64]. Meanwhile, the practical application will increase the ease of use. The application has been made more valuable in data management and processing to reduce data input and problem-reporting processes [27]. Regarding ease of use, productivity has been considered the primary motivation for the benefit of digital communication applications and health services. Chatbots can offer 24/7 digital support to perinatal women and their partners to make obtaining accurate, credible information more accessible, efficient, and intuitive than conventional means (e.g., books, internet searches, acquaintances, and healthcare professionals) [35]. Therefore, the operation of the question-and-answer application as a provider of information that answers semi-automatically can support application performance so that it can be “always active” and be used 24 h non-stop according to user needs.

(10)Instruction Clarity

The clarity of these instructions is expected to be a quality bridge for user needs. Thus, the interface program must be excellent by observing the interaction between the user and the mobile application, using graphical information or visual widgets, such as text boxes and clickable buttons. In this case, it is the correspondence of the submenu symbols between the symbols and their functions [7]. Understanding the language can also support the clarity of instructions in running applications to improve the readability of information [24,61]. Regarding service preferences, including the selection of service media and automatic service features, it is necessary to have clear instructions in operating along with design evaluations with planned features to prevent miscommunication between service providers and users [7].

(11)Language Usage

*Chatbots* are software-assisted intelligent services that communicate with people in their native language via voice or text. The most popular Natural Language Processing used in various enterprise end-user applications is generating answers to user requests in human-like natural language [55]. Therefore, in exploring the content of this menu, it is essential to examine what language will be used in programming the *chatbot*. In one study, it was said that all participants preferred to receive customized maternal health-related information on their mobile phones in the local language [61]. It should be expressed in simple language to improve the readability of information. It is also expected that the application menu will allow patients to switch between different languages, as in the qualitative finding that requires the use of bilingual language that can change automatically to learn how to communicate about symptoms [24]. The selection of greetings on digital communication menus and digital health services must be designed to direct users to find comfort when seeking health information and medical assistance for prenatal and postnatal care. The use of emojis in the *chat feature* and adjustment of voice tone must also be considered to improve professionalism while creating patient satisfaction and comfort in using the application menu [35].

(12)Substance

Continuing midwifery care can be provided through a continuous care model that provides a known midwife to follow women during pregnancy, birth, and the postnatal period (postpartum, infant and toddler, and family planning) for all women, both low- and high-risk and in all environments. The provision of midwifery care using this technology must also pay attention to the limits of its authority in providing health services in the form of valid and *evidence-based information*. *Because* many websites and most smartphone applications are unreliable sources of information, this is a weakness of current *digital health applications* [14,30].

The quality of the content and the expertise of the *chatbots* must be considered first in the development process, which may influence the intended use by the users [35,65]. Previous studies tried to expand the knowledge database of *chatbots-based question-and-answer (Q&A) applications* to ensure content quality and improve response capability using a validated web crawler. This program can automatically collect all accessible web pages so that users can quickly find the content they need [35,65]. The health content offered is created and validated by specialized professionals and supported by research institutions to increase the reliability of patients in obtaining information to maintain the quality of information substance content [35,60].

(13)Benefit

The World Economic Forum argues that the COVID-19 pandemic is an accelerator for *chatbot technology*, helping people around the world become more comfortable with utilizing these tools for healthcare as the adoption of *chatbots* in broader healthcare applications will continue to grow [66]. In terms of increasing the effectiveness of services, *digital health can* improve the health of mothers and babies in midwifery services, making it easier for health workers to reach and access patient data in real time, faster and more efficiently [8,25,67]. *mHealth technology* is feasible and can improve the detection of complications and timely referral to treatment facilities [31,68]. Clients are more comfortable using *mHealth* in seeking health information. It is because the provision of *mHealth-based applications is* calculated and prioritized to increase access to health services for high-risk pregnancies to improve emergency response and to address changes in patient habits and the way they seek accurate health information [13,69].

Behavioral and health habit changes can occur when health workers respond well and support digital communication in health services and want to use it in pregnancy monitoring. It also effective for promoting exclusive breastfeeding at an early age; this can also improve good relations between health workers and patients in terms of social change [69,70,71,72]. Moreover, *mHealth* can provide an acceptable and affordable alternative approach to maternal and child healthcare in face-to-face settings [61]. Thus, the application’s menu-based continuous midwifery care will allow more intense interaction and communication, thereby developing a close relationship between health workers and patients where women feel free to express their concerns and create a forum to share their experiences about pregnancy [53].

(14)Appropriate Value

Information can be easily understood by patients if it is a visual message played in their local language and can be played back and listened to as often as desired, such as information relayed via video and audio media [61]. Thus, in making a new application menu, it must also adapt to local culture and customs, one of which is the regional or local language. This is in line with research that states that if there is an application menu that allows more intense interaction and communication, it can develop a close relationship between health workers and patients so that they are more convincing in providing health information [53]. Then, success or failure of digital health implementation depends on the acceptance, motivation, and performance of midwives and their ability to adjust to the cultural and social environment of the local community [30,31].

(15)Supporting Components

Research on iPosyandu states that respondents expect information on the application to be available at any time [7,9]; thus, our research will design an *offline version that is* planned to have the capacity to store data on Android devices and then synchronize with the server when connectivity is available [9,63]. Data connectivity is necessary for operating *chatbot-based health applications*. For the long response time of *chatbots*, some of them consider problems in mobile data connectivity as the cause of the slow response because this process involves digital signal processing to convert the input speech into output text [43,57]. The data processing process places a considerable preparation requirement on the processor and server memory assets in the form of server processors and memory resources that contain extensive procedures. Each user uses the work structure input mechanism for signal generation, meaning it translates the signal to the user [44].

The data capacity for data use and storage must also be considered in the initiation *of telemidwifery*. In one study, it was said that the prototype was designed to operate at 2.4 GHz and 3.3 V. The transceiver supports a communication range between 20 m and 100 m with a transmission speed of 128 kB/s and a latency of 1.2 ms. *The transceiver* combines AES to encrypt data in *real time* and has a high common-mode rejection ratio (CMRR) [73]. Regarding the operation of the *telemidwifery menu*, the village midwives want an operator or customer service, such as an admin, who can manage *telemidwifery* apart from them. As in the study, it was stated that the functionalities in the form of indications when the user was online, push notifications, and read receipts were managed manually with the help of research staff [57]. We can also translate this assistance as an admin who can direct village midwives in providing continuity midwifery services via *telemidwifery*.

Then, from the 15 content menu findings, what’s the relationship between them? Thus, to provide a more precise link between each content menu, the effectiveness of the telemidwifery menu application is illustrated in Figure 4.

(16)ISO/IEC 9126. Attribute Finding

In this study, researchers found three characteristics of the ISO/IEC 9126 Model software quality development from the qualitative results: usability, *functionality*, and *efficiency* [74,75,76]. The concept of *reliability* was not found in the qualitative results because the qualitative research explored the expectations, hopes, desires, and preferences of village midwives and pregnant women. It is regarding the expected new application menu apart from the implementation of the application menu trial and not in the form of an evaluation test. According to the Informatics Expert, these three characteristics are sufficient for the initial development component of the new application menu.

In this study, the *usability subcategory includes* menu naming, *telemidwifery features*, clarity of instructions, ease of use, appearance, attractiveness, substance, use of language, suitability of values, service media, and benefits. Part of these findings differ from previous findings in that the subcategories are only understandability, ability to learn, operation, and attractiveness (Figure 5) [41,77].

*The functionality attributes based on semi-automatic chatbots* are digital communication, digital health services, and digital check features. In another study, the functional attributes included interpreting commands accurately, carrying out the requested task, interpreting knowledge, maintaining discussion, activation, the number of services available in the chatbot, suitability, accuracy, compliance, and security [42,77]. Meanwhile, some attributes are implicitly similar to previous research in the number of services available on *chatbots* in the subcategory of digital communication, digital health services, and digital service features (Figure 5). The attributes are expected to be able to interpret orders accurately, execute requested tasks, be flexible in interpreting knowledge, maintain discussion, and be easy to activate. Thus, the attribute findings in *telemidwifery* can be refined based on previous results after the application trial is carried out later.

In another study, it was said that high-level usability characteristics were described in subfactors such as global site understanding, online feedback and help features, interface and aesthetic features, and other features. The functional aspects are divided into search and retrieval problems, navigation and browsing problems, and student-oriented domain-related features. In this case, we can conclude that there are similar attributes between functionality and usability in the form of features that can support application operations [76]. Similar to the findings in the research on the initiation of *telemidwifery* menus, there are subcategories of *usability* and *functionality*: service media and benefits (Figure 5).

Subcategories related to efficiency in this study are supporting components that contain operational hours, signal conditions, data quantities, and menu operators. Similar to the results of other studies, attributes that are included in efficiency for *chatbot-based applications* are ease of use, fast response, availability at any time, accessibility, and need to take into account the ethics of time, resources, utilization, and adaptability to manipulation of input data by users [42,74]. Although the attributes are not the same as the previous research, the characteristics of operating hours are related to the fast response, time ethics, and availability at any time. Attributes of data magnitude and signal state can match accessibility, and resource attributes and menu operators can be associated with account and resource requirements. In contrast to the findings in this study, the ease-of-use attribute is not included in the efficiency category but the *usability category*. Meanwhile, the benefit attributes fall into the *usability* and *functionality categories* (Figure 5).

### Strengths and Limitation

The primary strength of the present study is the rare term “telemidwifery” in menu applications, which represents the identity and the goal of midwifery continuity-of-care services. The initiation of this menu explores the expectation, preferences, and points of view of the respondents to build and design the pre-development menu. However, the initiation of the telemidwifery menu, especially for the digital check feature, could not be built and implemented immediately because there were limited resources, and a longer time was required to adjust the menu for smartphone-based applications that were integrated with the internet of things (IoT) technology. The pre-development of this application menu has the opportunity to be developed in Indonesia because, currently, the Ministry of Health has developed a portable-based digital examination tool that is connected via Bluetooth or the cloud to automatically input the result into a smartphone.

## 5. Conclusions

The characteristics of the content in the telemidwifery menu in the development of an application for midwives are naming, digital communication, digital health services, telemidwifery features, digital examination features, service media, appearance, substance, attractiveness, convenience, benefits, clarity of instructions, use of language, suitability of values, and supporting components. Village midwives and pregnant women expected and preferred to operate a telemidwifery menu with semi-automatic chatbot features. These findings have also been compared to the software quality attributes from the ISO 9126 Model in the form of usability, functionality, and efficiency in the design of telemidwifery menu content. These three characteristics are sufficient for the initial development component of the new application menu. Testing the application menu will be a topic for further research because, at this research stage, it is only a menu prototype still in pre-development.

## Figures and Tables

**Figure 1 ijerph-19-10713-f001:**
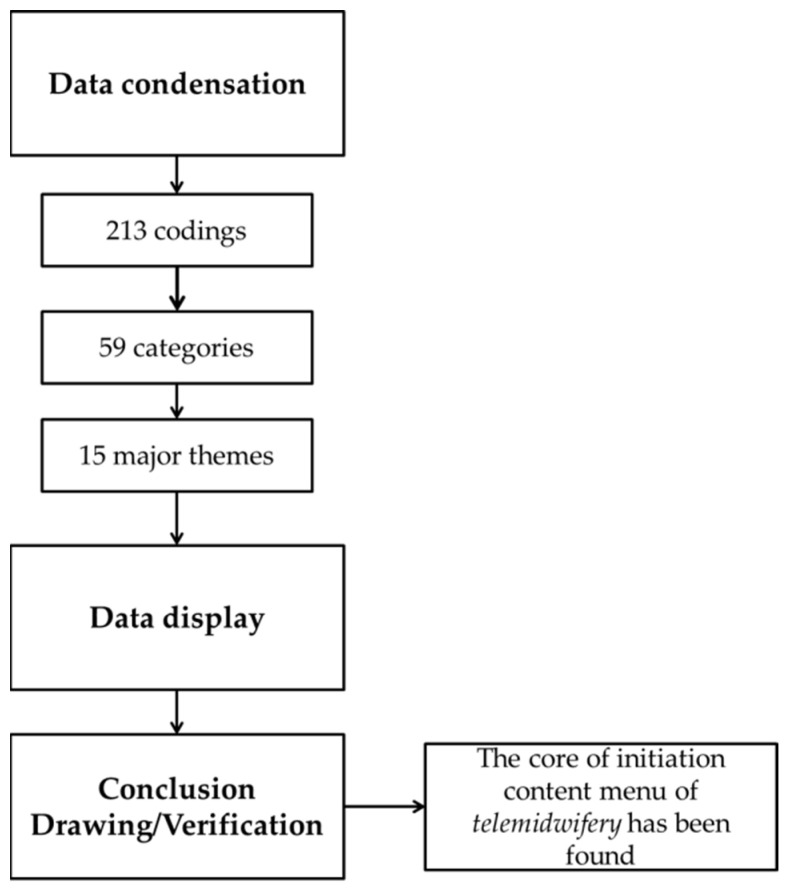
Qualitative Data Analysis Process.

**Figure 2 ijerph-19-10713-f002:**
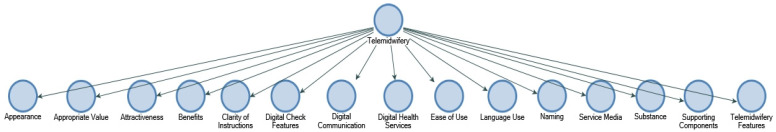
Visualization of Major Themes in Telemidwifery.

**Figure 3 ijerph-19-10713-f003:**
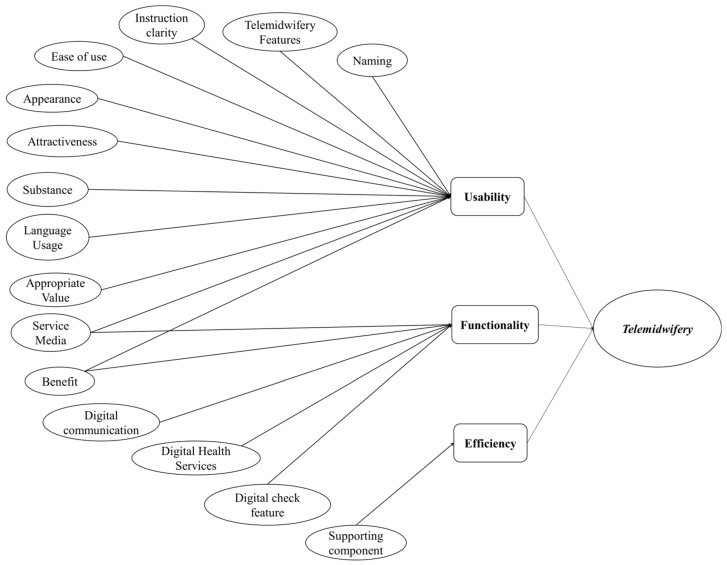
Telemidwifery Menu Content Concept.

**Figure 4 ijerph-19-10713-f004:**
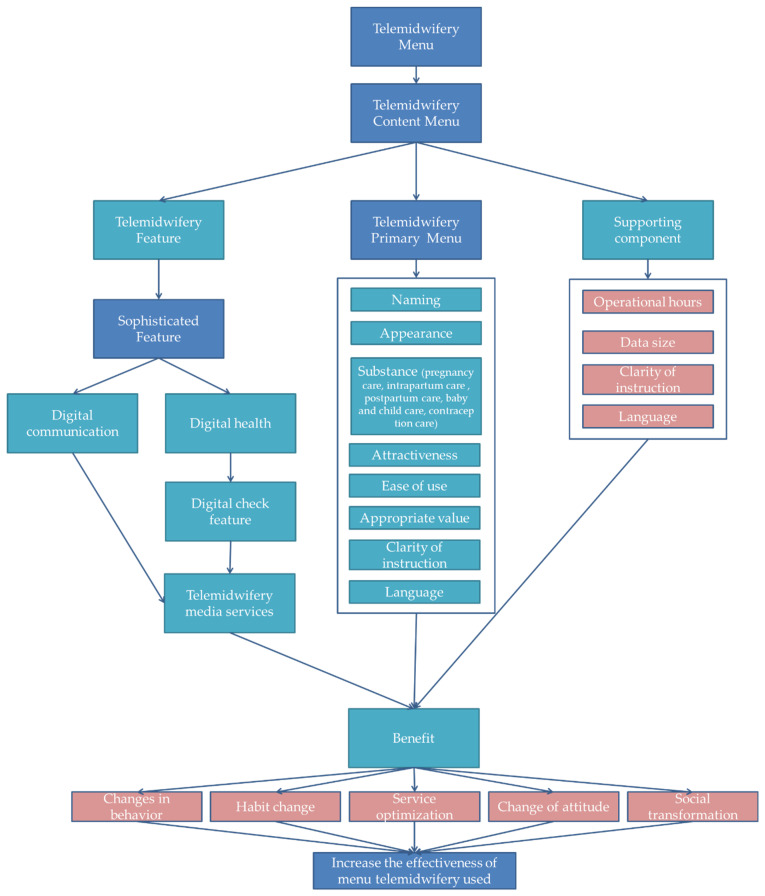
The illustration of the telemidwifery menu to prove its effectiveness in use (the blue color indicates the main menu concept, the aqua blue color is the characteristic menu content of telemidwifery, and the pink color is the supporting indicators of the menu).

**Figure 5 ijerph-19-10713-f005:**
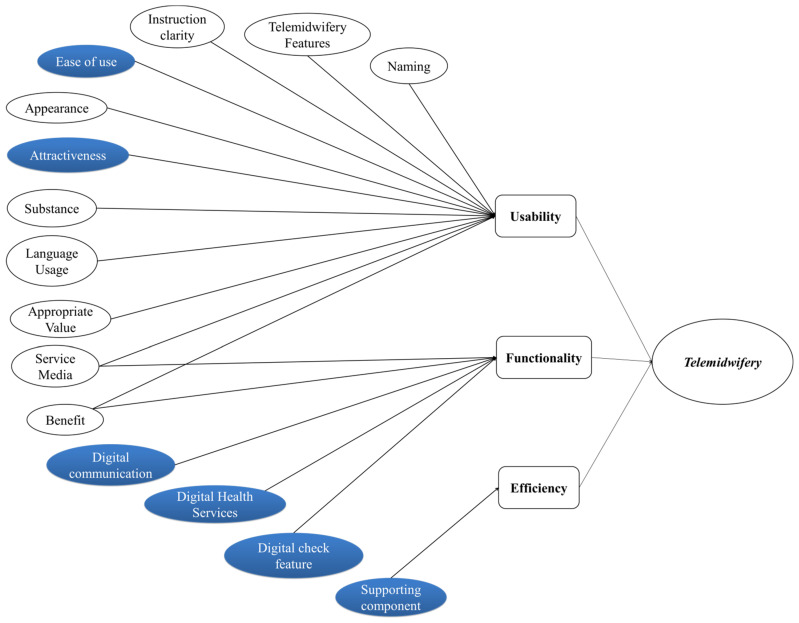
The comparison of Telemidwifery ISO/1926 attribute findings with previous research (common attributes marked with blue color).

**Table 1 ijerph-19-10713-t001:** Themes and Key Insights of Telemidwifery Menu.

No.	Theme	Key Insight	Quotations
1	Naming	i. Presenting the goal ii. Representing identity	*i.* *That’s enough because we have a telegram application. So, in the community, users may already know the term “tele” that provides information, like a telegram application (Informant 14).* *ii.* *So, Telemidwifery can represent the midwife’s identity (Informant 10). The title’s already using the term “midwifery”, so it has been highlighted (Informant 9)*
2	Digital Communication	i. Manual ii. Automatic	*i.* *We can ask the midwife directly (pregnant mother 1) so that we know more clearly (pregnant mother 2)* *ii.* *Yes, it’s better if it automatically answers (Informant 3). For example, if any question for the midwife, the application will answer (Informant 7)*
3	Digital Health Services	i. Manual ii. Automatic iii. Semi-automatic iv. Response period	*i.* *The robot will immediately comment or answer the question (Informant 15).* *ii.* *So, if we are unsatisfied, we can ask the midwife manually(Informant 15).* *iii.* *That’s why it was the term “tele” and “semi-automatic” that was mentioned earlier (Informant 15)* *iv.* *Eee, I can’t answer at once, because we have activities (Informant 15), we have a lot of work burden (Informant 13) give the waiting time (Informant 14), Eee, 5 min, 10 min, 30 min is the maximum, Ma’a (Informant 10). If possible, don’t take too long waiting time, 15 min, 10 min (Pregnant woman 3)*
4	Telemidwifery Features	i. Other menu links ii. Notifications iii. Reminder iv. Reschedule v. Translator vi. Zoom text or image	*i.* *The doctor’s consultation. For example, we get a case that is not under our authority. We can immediately consult a doctor. (Informant 3)* *ii.* *Yes, it means notification if there is an incoming message with certain information (Informant 11)* *iii.* *Yes, that’s how I was reminded (Informant 12), for example, the reminder about the current and following visiting schedule. It was like that reminded the day before scheduled (Informant 11)* *iv.* *For instance, it’s like this: what time are you at home? For example, there’s a sudden meeting when it’s on the schedule. Later, the date will be rescheduled for a few days or a few hours (Informant 13)* *v.* *We need (Informant 13), we can translate anyway (Informant 12), because most of us have difficulties in Indonesian. Emm sorry, I mean in English (Informant 15)* *vi.* *Well, yes, it may have a ‘zoom’ feature. What is the name of an application that can magnify the feature, maybe? (Informant 15), just automatically.(Informant 12), everything is zoomed in.(Informant 11)*
5	Digital Check Feature	i. Digital sensor tool History Physical examination Supporting investigation Vital signs check ii. Digital checking accuracy	*i.* *Blood pressure; maybe the baby moves in a Leopold using remote care to know the location of the fetus and detect early complications from its position (Informant 10. Yes, at least it’s like that, the fetal heart rate is the same as the body weight. If measuring the body weight is not possible yet, yeah? (Pregnant 5). Heart rate detection, because sometimes it slow in rate. If the speed is good like this, how much is the heart rate score suitable?* *ii.* *I doubt the accuracy. Well, with a trial, we can make it possible, whether the accuracy is accurate or not (Informant 14)*
6	Service Media	i. Written media ii. Visual media iii. Audio–visual media iv. Audio media	*i.* *It means chat (Informant 11), chat is available (Informant 15), at least she chats more often (Informant 11)* *ii.* *It means photo or image, send a picture (informant 2)* *iii.* *Video call, emmm video call, basically all supported media. It’s all like a WhatsApp application. Video call (pregnant woman 3), it’s better to talk face to face (pregnant woman 5);* *iv.* *We can send voice notes, pictures, and video calls (Informant 3). Just the voice rather than voice note, best to do telephone. (Informant 15) Voice notes are okay (pregnant mother 1) So that we don’t get tired of typing (pregnant mother 6)*
7	Attractiveness	i. Visualization ii. Appearance iii. Service delivery iv. Credibility v. Transportation availability vi. Completeness vii. Feature viii. Offline access	*i.* *Can a panthom props (Informant 12), an animated picture, visualize it as a graphic? (Informant 13)* *ii.* *If the application is interesting, it excites the patient to use it (Informant 9). Exciting applications and menus that contain many pictures make it not bored to read (Informant 9)* *iii.* *Yes, it’s better if someone answers it automatically, ma’am (Informant 3); if robots can help, it’s better if they can do everything, ee which is called “semi” at the beginning (Informant 15)* *iv.* *So let the community or pregnant women know why we do not provide such specific medical intervention. Not because we don’t want to do it, but we have an eeee referring to the Minister of Health rules (Informant 14)* *v.* *Not all of them have vehicles, ma’am. It’s a pity that there isn’t one, so it isn’t effortless. Especially for the patient far away from health facilities also an emergency. Yeah, just according to the application (Pregnant women 2)* *vi.* *It’s better if it is complete (Informant 15)* *vii.* *So it’s like a filter (Informant 14), interactive, and communicative (Informant 13)* *viii.* *Yes, getting offline access is even better (pregnant mother 5). I want to get the answer immediately if possible (Informant 3)*
8	Appearance	i. Menu icon ii. Bookmarks iii. Menu content writing iv. Display color v. Text color	*i.* *Only the midwife icon (Informant 8) holds the phone, then there is no icon for the baby and pregnant woman, but it more shows us, the midwife (Informant 14), what sign is there, the phone or something (Informant 11), Yupss, it’s a “tele”, right? The midwife while holding a cellphone or a stethoscope (Informant 11)* *ii.* *The usual one is black color, so the critical thing is with red marks to see the difference meaning (pregnant woman 1), it’s italic or bold, red color marking (Informant 8), the important thing is only (Informant 15)* *iii.* *I don’t know the type of font. But the font can attract the attention of patients to use it. So it’s not rigid and adapted to current development fonts. (Informant 10)* *iv.* *It’s preferred pink color, which is the same color as a pomegranate (Informant 12); the MCH book is also pink (Informant 11)* *v.* *It’s black because it’s the colored background, so if the writing is colorful, it will be difficult to read (Informant 4)*
9	Ease of Use	i. Sophistication ii. Practical iii. Unlimited access	*i.* *⋯ or more sophisticated (Informant 14)* *ii.* *So, we don’t need the complicated feature but an easier one. Anything simple, practical, and easy (Informant 5)* *iii.* *24 h, operational hours (Informant 8), and almost 12 h from morning to night (Informant 14)*
10	Instruction Clarity	i. Submenu symbol compatibility ii. Language comprehension iii. Service preferences	*i.* *For example, if we want to operate telemidwifery menus, There is a notification. If we click on the menu, the video feature menu will appear in the various menu. Yeah, the submenu (Informant 5)* *ii.* *Maybe we need a sub-bilingual language (Informant 14). Yeah, there are two languages (Informant 13), just like this, there are two, so when we open it in Indonesian, we understand what they want and talking about (Informant 14).* *iii.* *Is there provide a choice for us? The intention of the provider. For example, if now, we can’t respond to a question using a video call or telephone, we want to respond using chat. (Informant 14)*
11	Language Usage	i. Delivery method ii. Language type iii. Greeting	*i.* *So, ma’am, for the chatbot is using formal language type (Informant 15), but for the manual stage we can use formal and informal language type (Informant 15),* *ii.* *Because not all of them can speak Sundanese, Indonesian is the national language (Informant 1),* *iii.* *But maybe it’s just a general greeting because we’re in Indonesia. Here, there are various religions. So use the standard greetings (Informant 15), hello (Informant 12)*
12	Substance	i. Pregnancy care ii. Delivery care iii. Postpartum care iv. Baby and toddler care v. Family planning care vi. Reference sources	*i.* *The hemorrhaging case must be answered quickly. Those are the danger signs (Informant 14). It’s a diet problem. Sometimes I’m afraid I’ll eat the wrong food (Pregnant woman 5). We can also use it to find medicine because over-the-counter drugs are not allowed. (pregnant woman 4)* *ii.* *Early signs of labor (Informant 9), counting the contraction waves (Informant 8),* *iii.* *Postpartum care, exclusive breastfeeding, and breast care so that there are no blisters (Informant 9)* *iv.* *Complementary feeding, yeah, complementary feeding menu (Informant 14), umbilical cord care, and the stage of immunization is vital (Informant 5)* *v.* *Postpartum contraception planning (Informant 9). The period to using contraception (pregnant woman 3)* *vi.* *The fundamental law (UUD), the Minister of Health rules (Informant 14), Journal, If the journal is more updated (Informant 10). From a book, maybe from a book (Informant 14), I think it’s more like the mother and child health (MCH) book (Informant 13); I believe it’s more complete than others (Informant 13)*
13	Benefit	i. Service optimization Complications—early detection Service effectiveness Service speed Conformity of authority Emergency response ii. Habit change iii. Changes in behavior iv. Change of attitude v. Social transformation	*i.* *Blood pressure, maybe if the baby moves, perhaps the baby is in a leopold remote care. It aims to know the position of the fetus, and complications from its location (Informant 10). In terms of time, because we are on working and can’t come at once (Informant 14), more efficient (Informant 11) costs, reducing transportation costs (Informant 14). So, if we already know that, we already have the application, the visit can pass like that, so the checking can be easier. So there is no need to come to the place (Informant 5),* *ii.* *Yes, she can handle it by herself (Informant 13), and she can practice it at home (Informant 14)* *iii.* *So, so that the mindset can be changed to be painstaking, so it can be checked (Informant 2), improve the health of the mother and child in essence (Informant 15),* *iv.* *It’s more like increasing the knowledge, so then not curious (pregnant mother 1)* *v.* *Maybe it can increase closeness because they often interact (Informant 9)*
14	Appropriate Value	i. Culture value ii. Social value	*i.* *Here, the custom is that they’re afraid their pregnancy will fail. So, we check sometimes it’s been four months already. Even though it should have been checked three times. (Informant 3)* *ii.* *Maybe because the patient already cares for us, it is possible to increase the closeness due to frequent interactions. It adds a more internal familiarity. (Informant 9)*
15	Supporting Component	i. Data size ii. Operational hour iii. Signal requirement Signal state Area conditions Network connectivity iv. Menu operators	*i.* *If it’s not big, it doesn’t matter. If it reaches gigantic, it’s quite troublesome (Informant 14)* *ii.* *Operational hours (Informant 13), there may be appointed time like that (Informant 12), so we have to stand by, for example, from this hour to this hour (Informant 12), yes, if we can schedule it like that, it provide almost be 12 h services from morning to night, right? I mean, it’s until 8 p.m. (Informant 14)* *iii.* *Problems because it is in urban areas. So the connectivity of the signal is strong, different in countryside areas (Informant 10). Sometimes the connectivity is lost (pregnant woman 1); sometimes, when it rains (pregnant woman 1), it usually depends on the particular location (pregnant woman 2)* *iv.* *There must be a particular operator; if not ⋯ (Informant 14), It’s more convenient if managed by one person (Informant 13); yes, it’s like a call centre (Informant 14)*

## Data Availability

Not Applicable.

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
