# Peer review of "Exploration of Telemidwifery: An Initiation of Application Menu in Indonesia"

_ijerph, 2022, doi:10.3390/ijerph191710713_

Round 1

Reviewer 1 Report

Thank you for inviting me to revise this paper titled Exploration of tele midwifery: an initiation of application menu in Indonesia. The study concerned the design of an application (iPosyandou) to overcome communication barriers and difficulties, and thus improve the midwifery services in rural areas. Authors combined the perspective of midwives and pregnant women. By evidencing the role of telemedicine and ICT application in the health industry, this study is likely to have an impact on community health (namely reduce the maternal and infant deaths) in rural areas. The study is quite well structured, but some major revisions are needed. Please find in the following my suggestions to improve the paper.

In the introduction, I would suggest authors to rephrase the research aim. In the actual draft, there is a long sentence that makes difficult to follow the underlying reasoning. I would also invite authors to consider the role of social networking sites in the health care industry that are increasingly used by institutions, health-care organizations and users (patients and not patients) for a number of reasons and scopes as explained in this review Pianese, T., & Belfiore, P. (2021). Exploring the social networks’ use in the health-care industry: a multi-level analysis. International Journal of Environmental Research and Public Health18(14), 7295.

As for Methodology, I would invite authors to rephrase the study design section and include the information about research ethics at the end of the paper (check if the journal has a specific section for this information). Likewise, the purposive theoretical sampling technique should be mentioned in the starting of the paragraph, as you used this criterion for both midwives and pregnant women’s selection. Second, authors should provide with more information about FDGs discussion: who led the conversation, did you explain the aim of the research to the participants, did you established some general topics to be discussed (those included in the appendix B. if yes, you should mention in the text)? For example, in focus group there is a “moderator”, i.e. someone that is in charge of leading the conversation (e.g. avoid to discuss about off-topic arguments). Third, authors should provide with more information about coding activity: e.g. did you follow an inductive or deductive approach in coding process? Fourth, the limited number of participants should be included among the limitations of the study in the conclusive section. If you selected 22 midwives and 11 pregnant women, why did only 15 and 6 took part to the FDG discussions?

As for Results, I would suggest to include the key results in the table 1, along with examples/quotations. In my opinion, this would help readers in immediately realize the main findings emerging from your study. E.g. for the second theme, in the table you could report that pregnant women and midwives would prefer a mix of manual-automatic response depended on the urgency (and thus on the need to have a Synchronous/asynchronous responses) and the ability to elaborate a standardized question etc. But this is up to the authors to decide if this way of presenting results could be effective or not.

Then, it seemed there is an overlap between the theme digital communication and digital health services: do you think it would be effective to aggregate them or to better explain the differences between them? I also invite authors to maintain only the relevant quotes (e.g. I do not think it is really necessary for “naming” because midwives identity is not really the core of your research).

Please make clearer the link between design features for the effectiveness of the telemidwivery application (easy to use, language, color etc) and the cultural and social value. Your focus is on technical issues or on behavioural issues? Finally, the conceptual framework arrives quite unexpectedly, without ever having been mentioned in the course of the study. I don't know if it makes much sense, it's about some general ISO categories to which it leads those (inductively or deductively) that you identified from the group discussions.

Finally, the discussion is really too rich and repeat many arguments already discussed in the results sections. I would ask authors a consistent effort to evidence the theoretical contribution of their study, i.e. the way this study enriched existing knowledge on the topic.

Hope these comments will help you.

good luck with your paper!

Author Response

Thank you for your advice. I have revised the manuscript based on your suggestion. The detailed response to your comments will include in the attachment below.

Thank you for your advice and awaiting your prompt response.

Reviewer 2 Report

Thank you for the opportunity to do this review. This article describes research exploring the telemidwifery menu content for village midwives and pregnant women in the Purwakarta Regency, West Java, Indonesia. I think the research performed is interesting, but some minor aspects could be revised. These are the following.

- The introduction is interesting but also a bit long. Even more, I believe that the justification of the study, albeit shown, could be explained better. The aims of the study perhaps could be described better. Finally, I do not understand the sentence «Sepanjang pengetahuan penelitian istilah telemidwifery jarang dipakai». It seems an Indonesian sentence means, «To the best of the research's knowledge, the term telemidwifery is rarely used». I understand that text is superfluous there.

- In this research, potential selection bias is paramount due to the low number of final participants. Thus, I would invite the authors to describe the eligible population's main features better. I would also propose that they describe some specific aspects of the recruitment process better. For example, how the potential participants were chosen, why they use «with a high and a low number of maternal and infant deaths» as a criterion, or how they define a high and low number of maternal and infant deaths. I would invite them to explain these aspects, among others.

- I would propose that the authors not only describe the potential selection bias but also assess it regarding the external validity of their conclusions.

- I would invite the authors to explain better the criteria «Pregnant women in trimesters 1-3». I understand this means all pregnant women?

- The results and the discussion sections are also interesting. However, these sections are long and, sometimes, a bit confusing to read. If possible, again, I would invite the authors to shorten them. This is just a proposal that I believe could improve the overall readability of the article.

- I would invite the authors to describe the potential limitations of the research in a specific section and, if possible, assess them regarding the external validity of their findings.

- The conclusions perhaps could also be shortened.

Author Response

(The authors gave the same response as above.)

Reviewer 3 Report

The work titled 'Exploration of Telemidwifery: An Initiation of Application Menu in Indonesia'. the work is not acceptable in its present form and must revise before resubmission.

1. Please place those same "topic" while you review the literatures. I would suggest you to restructure your literature review section. Current form is hard to read.  To be legible, the whole text must be completely edited with the help of a native English editor to polish your writing

 2. I would suggest the author to re-organize this manuscript to readable structure. For instance, the method should place those sections in the context. The experiment result needs to place to those results.

3. Discussion section is hard to read/understand. Too long. Seems like Literature review.

4. authors should enhance their findings, limitations, underscore the scientific value added of your paper, and/or the applicability of contributions/shortages and future study in this session

Author Response

(The authors gave the same response as above.)

Reviewer 4 Report

This paper discusses the problem of the midwifery continuity of care model. Many helpful suggestions are given. However, the article lacks readability, lacks diagrams due to too much text, and the only two diagrams are not of high quality. It is recommended to further sort out the content of the article and increase the representation of graphics and tables. The discussion part is suggested to be expressed in subsections or (1)(2).... The current content is difficult to understand quickly.

Author Response

(The authors gave the same response as above.)

Round 2

Reviewer 3 Report

Authors have addressed all the revisions/ corrections. Now arctic;e can be accepted in its present form.

Author Response

Thank you very much for your advice. Hopefully, this manuscript deserves acceptance in this journal.

Reviewer 4 Report

Fine revision.

Author Response

(The authors gave the same response as above.)
